# Phenotypic Alteration of BMDM In Vitro Using Small Interfering RNA

**DOI:** 10.3390/cells11162498

**Published:** 2022-08-11

**Authors:** Noreen Halimani, Mikhail Nesterchuk, Irina N. Andreichenko, Alexandra A. Tsitrina, Andrey Elchaninov, Anastasia Lokhonina, Timur Fatkhudinov, Nataliya O. Dashenkova, Vera Brezgina, Timofei S. Zatsepin, Arsen S. Mikaelyan, Yuri V. Kotelevtsev

**Affiliations:** 1Vladimir Zelman Center for Neurobiology and Brain Rehabilitation and Center of Life Sciences, Skolkovo Institute of Science and Technology, Moscow 143025, Russia; 2Koltzov Institute of Developmental Biology of Russian Academy of Sciences, 26 Vavilov Street, Moscow 119334, Russia; 3National Medical Research Center for Obstetrics, Gynecology and Perinatology Named after Academician V.I. Kulakov, Russian Federation, 4 Oparina Street, Moscow 117997, Russia; 4Department of Histology, Pirogov Russian National Research Medical University, Ministry of Healthcare of The Russian Federation, 1 Ostrovitianov Street, Moscow 117997, Russia; 5Department of Histology, Cytology and Embryology, Peoples’ Friendship University of Russia, 6 Miklukho-Maklaya Street, Moscow 117198, Russia; 6Scientific Research Institute of Human Morphology, 3 Tsurupa Street, Moscow 117418, Russia

**Keywords:** macrophages, polarization, siRNA, IRF5, IRF3, EGR2, TLR4

## Abstract

Autologous macrophage transfer is an emerging platform for cell therapy. It is anticipated that conventional macrophage reprogramming based on ex vivo polarization using cytokines and ligands of TLRs may enhance the therapeutic effect. We describe an alternative approach based on small interfering RNA (siRNA) knockdown of selected molecular cues of macrophage polarization, namely EGR2, IRF3, IRF5, and TLR4 in Raw264.7 monocyte/macrophage cell line and mouse-bone-marrow-derived macrophages (BMDMs). The impact of IRF5 knockdown was most pronounced, curtailing the expression of other inflammatory mediators such as IL-6 and NOS2, especially in M1-polarized macrophages. Contrary to IRF5, EGR2 knockdown potentiated M1-associated markers while altogether abolishing M2 marker expression, which is indicative of the principal role of EGR2 in the maintenance of alternative phenotypes. IRF3 knockdown suppressed M1 polarization but upregulated Arg 1, a canonical marker of alternative polarization in M1 macrophages. As anticipated, the knockdown of TLR4 also attenuated the M1 phenotype but, akin to IRF3, significantly induced Arginase 1 in M0 and M1, driving the phenotype towards M2. This study validates RNAi as a viable option for the alteration and maintenance of macrophage phenotypes.

## 1. Introduction

Recent progress in CAR-T therapy warrants further developments based on the modification and transplantation of myeloid cells for the treatment of cancer, liver and lung fibrosis, and many other pathologies [1,2,3].

The feasibility and efficacy of allogeneic bone-marrow-derived macrophage (BMDM) transplantation were demonstrated in animal models of pulmonary alveolar proteinosis [4], visceral leishmaniasis [5], and liver and idiopathic pulmonary fibrosis [6,7].

The first phase one in-human trial of autologous macrophage transplantation in patients with cirrhosis has already been completed successfully. Each subject received a single peripheral injection containing 10^7^, 10^8^, or up to 10^9^ cells. The procedure did not cause any dose-limiting toxicities, transfusion reactions, or macrophage-activation syndrome [7].

Infiltrating or tissue-resident macrophages play a central role in many pathological processes, including tissue injury and repair, as well as in immune tolerance and resolution of inflammation [8]. Exhaustion of tissue-resident macrophages and the imbalance of proinflammatory (M1) and pro-regenerative (M2) phenotypes constitute important pathologic conditions in many disorders [9,10,11,12].

Macrophages provide an ideal platform for cell therapy. They are easily accessible through plasmapheresis or via differentiation from bone-marrow-derived progenitor cells. Macrophage progenitors and differentiated macrophages can be transfected with expression vectors, synthetic mRNA, or oligonucleotides. Existing strategies for macrophage reprogramming using cytokine cocktails or ligands of LPS receptors, RNAi knockdown, and genome editing, and in vitro-transcribed RNA overexpression was recently reviewed [13]. It is anticipated that ex vivo or in vivo macrophage reprogramming into proinflammatory or regenerative phenotypes can be essential for treating specific disease conditions associated with the imbalance of M1/M2 polarization.

Here, we present RNAi-based protocols which complement ligand-based genome editing and synthetic mRNA expression techniques for in vitro macrophage reprogramming. RNAi technology can be applied to any target gene, does not affect the genome, does not persist after cessation of treatment, and at the same time is sufficiently long-lasting to have a significant effect in vivo.

Evaluation and validation of targets for macrophage reprogramming have become a central field of biomedical research which employs cell culture experiments, NGS, and bioinformatic analysis [12,13,14]. This paper investigates the effects of RNAi-based knockdown of several carefully selected target genes, including early-growth-response factor 2 (EGR2), interferon-regulatory factor 5 (IRF5), interferon-regulatory factor 3 (IRF3), and Toll-like receptor 4 (TLR4) in the Raw264.7 cell line, as well as exploring the effects of knockdown on M1- and M2-associated markers of polarisation in BMDM.

## 2. Materials and Methods

### 2.1. Isolation and Culture of BMDM

Four-week-old FVB mice were humanely sacrificed according to the internal ethics regulations of the Koltzov Institute of Developmental Biology, Russian Academy of Sciences. The method of isolation and culture was performed according to the protocol of Ying et al. [15]. Briefly, the hind legs were cut off from the rest of the body without breaking the bones, i.e., the femur and tibia. The bones were stripped of skin tendons and cartilage using tweezers and then transferred into a 15 mL falcon tube containing sterile PBS. Working in the culture hood, both bones were cut off from both ends and then flushed with sterile PBS in a 26-gauge needle. The progenitor cells were centrifuged at 1100 rpm for 5 min. The supernatant was discarded, and cells were resuspended in DMEM F12 (Thermo Fischer Scientific, Waltham, MA, USA) supplemented with 10% FBS (Thermo Fisher Scientific, Waltham, MA, USA), 1% GlutaMax and 1% penicillin–streptomycin (Thermo Fisher Scientific, Waltham, MA, USA). Cells were seeded on a 10 cm, nontreated, sterile cell culture dish (Eppendorf, Hamburg, Germany), and 10 ng/mL of M-CSF (Miltenyi Biotec, Bergisch Gladbach, Germany) was added to induce differentiation of myeloid progenitors into mature macrophages. Cells were cultured at 37 °C and 5% CO_2_ for seven days and then harvested for subsequent experiments. At this point, M-CSF was removed from the incubation media.

### 2.2. SiRNA Transfection of BMDM and Polarization

We analysed two types of preparations. In protocol 1, cells were transfected and then polarized. In protocol 2, cells were polarized first, then transfected and then repolarized. In protocol 1, 2 × 10^5^ BMDM were seeded per well in 24-well plates 24 h before transfection. After 24 h had elapsed, the media was replaced, and BMDMs were transfected with 10 nM siRNA final concentration following the Lipofectamine RNAi Max (Invitrogen, Waltham, MA, USA) protocol. The cells were allowed to internalize the siRNA for 48 h in Opti-Mem (Thermo Fischer Scientific, Waltham, MA, USA), after which the media was replaced by media containing polarizing factors for 24 h, as described below. siRNA sequences for target genes are highlighted in Appendix A.

In protocol 2, equal amounts of BMDM were seeded and allowed to attach to the culture plate for 24 h, and then polarization ensued for 24 h. After polarization, the cells were transfected with siRNA as above. A duration of 24 h following transfection, the media was refreshed, and cells were repolarized under standard conditions. Repolarization also ensued for 24 h. The control group was transfected with siRNA for luciferase.

*Polarization:* For M1 polarization, BMDM were incubated in 24-well cell culture plates (Eppendorf, Hamburg, Germany) in complete growth media with 100 ng/mL *E. coli* LPS (Sigma-Aldrich, St. Louis, MO, USA) plus 20 ng/mL interferon–gamma (Miltenyi Biotec, Bergisch Gladbach, Germany). For M2 polarization, BMDM were incubated in a 20 ng/mL cocktail of interleukin-4 and interleukin-10 (Miltenyi Biotec, Bergisch Gladbach, Germany) in the same media. Polarization was allowed to proceed for 24 h before collection for analysis. The control group, M0, was maintained in complete growth media.

### 2.3. QPCR

Total RNA was extracted using ExtractRNA (Evrogen, Moscow, Russia). A measure of 1 μg RNA was reverse-transcribed to cDNA using the QuantiTect Reverse Transcription Kit (Qiagen, Hilden, Germany), and quantitative RT-PCR was performed using an Applied Biosystems StepONE Plus Real-Time PCR System (Thermo Fisher Scientific, Waltham, MA, USA) and qPCR mix-HS SYBR master mix containing SYBR Green I dye (Evrogen, Moscow, Russia). The mRNA levels of genes of interest were quantified via the ΔΔCt relative quantification method proposed by [16], with GAPDH as a housekeeping reference target. A primer list is provided in the Appendix A.

### 2.4. Western Blot

Protein was extracted using RIPA lysis buffer containing protease inhibitors (Thermo Fisher Scientific, Waltham, MA, USA). Measurement of total protein concentration was performed using a bicinchoninic acid assay. A measure of 25 ug of protein was loaded in each well, separated by 10% polyacrylamide gel electrophoresis, and then transferred onto a 0.45 µm nitrocellulose membrane. The primary antibodies used were specific for IRF3 (1:2000, Abcam, Cambridge, UK) and β-Actin (1:5000 Sigma-Aldrich, St. Louis, MO, USA). HRP-conjugated anti-mouse or anti-rabbit IgG (1:5000, Invitrogen, Waltham, MA, USA) were used as secondary antibodies. The reactive bands were detected using SuperSignal West Pico PLUS Chemiluminescent Substrate (Thermo Fisher Scientific, Waltham, MA, USA). Signal intensities were quantified using the Vilber Lourmat Fusion Solo S imaging system (Vilber, Collégien, France) quantitation software and normalized to β-actin as an internal control.

### 2.5. Flow Cytometry

All data were acquired using a Cytomics FC 500 flow cytometer operating on CXP software equipped with a dual 488 nm/635 nm (blue/red) laser (Beckman Coulter, Brea, CA, USA). Intracellular staining and staining of cell surface antigens were performed simultaneously using the MACS Inside Stain Kit (Miltenyi Biotec, Bergisch Gladbach, Germany). Briefly, Inside Fix buffer was added to 1 × 10^5^ cells and then resuspended in PBS. The solution was incubated at room temperature and pressure for twenty minutes, after which the cells were washed twice with PBS via centrifugation for five minutes at 300 g. The supernatant was discarded, and the pellet was resuspended in 100 µL of Inside Perm buffer; cells were stained with fluorochrome-conjugated antibodies at appropriate dilutions. Cells were incubated for fifteen minutes at room temperature and then washed with Inside Perm buffer. The supernatant was discarded, the pellet was resuspended in 400 µL of PBS, then flow cytometric analysis was performed. Antibodies against CD11b-VioBright FITC, CD86-PE-Vio770, and CD68-FITC were obtained from Miltenyi Biotec, (Bergisch Gladbach, Germany), whilst antibodies against CD206-PE and CD163-PE were obtained from eBioscience (San Diego, CA, USA).

### 2.6. Statistical Analysis

All the data points were included in the analyses, and no outliers were excluded from calculations of means or statistical significance. Ordinary one-way ANOVA was used to determine a statistically significant difference between the previously calculated means. A *p*-value of less than 0.05 was considered statistically significant for all comparisons. Statistical differences between different populations were determined by the recommended multiple-comparison post hoc test in GraphPad Prism 9.0 (GraphPad Software, Inc., San Diego, CA, USA).

## 3. Results

Monocytes isolated from bone marrow differentiated towards BMDM in 7 days under the influence of 10 ng/mL of M-CSF. The resulting BMDMs were characterized by the expression of macrophage-specific markers using flow cytometry and qPCR (see Appendix A).

### 3.1. Polarization of BMDM with LPS and IL-4 and IL-10

BMDM were replated and split into three equal parts designated for M1 and M2 polarization or nontreated M0. After treatment for 24 h, BMDMs acquired an M1 or M2 polarization phenotype, which was confirmed by measuring the expression of NOS2, IL-6, and TNF-α (M1 markers), and ARG1 and EGR2 (M2 markers). The relative mRNA expression of target genes, i.e., EGR2, TLR4, IRF3, and IRF5, in polarized macrophages, were recorded in M0-, M1-, and M2-polarized BMDMs and are presented in Figure 1 The expression of reference genes in M0 was taken as a baseline for comparison. Standard induction of NOS2 expression in M1 and ARG1 expression in M2 are presented in Figure 1a,b.

Polarization was initially checked using Raw 264.7 cells, see Appendix A. BMDMs react on incubation with LPS with an exponential increase in NOS2 activity, while incubation with IL-4 and IL-10 stimulated ARG1 expression. TLR4 and EGR2 were upregulated 5 times and 40 times, respectively, in the M2-polarized state. At the protein level, the costimulatory molecule CD86 is more abundant in M1 than in M0 and M2, whilst the mannose macrophage receptor CD206 is more abundant in M2 than in M1 and M0 macrophages, as shown in Figure 2.

### 3.2. Effect of EGR2 Knockdown on BMDM Polarization

siRNA suppresses EGR2 expression in M0 BMDM by 80%. This effect is strongly enhanced by LPS, where expression drops below 5% of the average level. However, suppression of EGR2 expression after powerful stimulation with IL-4 and IL-10 was relatively moderate. Out of the two EGR2 siRNA sequences we used, i.e., siRNA-6 and siRNA-13, siRNA 13, which showed knockdown effects in polarized Raw 264 cells, was not effective at all in polarized BMDM (see Appendix A).

Knockdown of EGR2 resulted in significant downregulation of ARG1 in M0 and M1 macrophages, where expression of the enzyme was already low. However, it had little effect on high-level ARG1 expression stimulated by IL-4 and lL-l0 in M2-polarized cells.

Markers of M1 polarization were upregulated by EGR2 knockdown. TNF-α expression was upregulated by EGR2 knockdown in all macrophage states, with maximum relative induction in M0 macrophages, where expression was ten times lower than in the M1-stimulated state. NOS2 was moderately upregulated in the M1-stimulated state and in the basal M2-stimulated state and, surprisingly, downregulated in M0 cells. IL-6, also a marker of M1 polarization, was moderately suppressed in the M1-stimulated state, increased in the M2 basal state, and was not affected in M0. Expression of IL-4 was relatively constant in all macrophage subsets, as shown in Figure 3.

### 3.3. Effect of TLR4 Knockdown on BMDM Polarization

We found that the expression of TLR4, a critical regulator of macrophage immune response and an activator of proinflammatory cytokines associated with M1 polarization, was upregulated by incubation with IL-4 and IL-10, with the cytokines inducing alternative M2 polarization. LPS, the natural ligand of TLR4, moderately downregulated the expression of TLR4 mRNA (Figure 1e). BMDM transfection with siRNA effectively suppressed TLR mRNA expression in the M0 state and M2 after IL-4 and IL-10 stimulation. Knockdown of TLR4 in the presence of LPS, on the contrary, was quite moderate in BMDM (Figure 4a). It is noteworthy that, in microglia isolated from newborn mice, knockdown of TLR4 mRNA was equally effective in M0, M1, and M2 states (Appendix A). Surprisingly, in all three polarization classes, M0, M1, and M2, the knockdown of TLR4 efficiently downregulated ARG1 expression, the primary marker of M2 polarization and a key enzyme in the metabolism of arginine, reducing the availability of the substrate for NOS2 (Figure 5a). Expression of ARG1 mRNA was suppressed by 80% even in M2 cells, where stimulated ARG1 expression was extremely high (Figure 1d). NOS2 was predictively downregulated in M0, M1, and M2 (Figure 4d). IL-6 was upregulated by TLR4 knockdown in the M1 group, in the presence of LPS, its natural ligand, and moderately but significantly downregulated in the M2 group. Both IRF3 and IRF5 had a similar pattern of moderate upregulation in M0 and M2 and significant upregulation in M1, with an even bigger relative effect than for IL-6 upregulation by TLR4 knockdown in the M1 class (Figure 4c–f).

If transfection with siRNA preceded polarization with LPS or IL-4 and IL-10 (protocol 1), the effects were quite different. Knockdown was effective in all states; however, knockdown of TLR4 had no impact on EGR2. M1-associated changes in M1 macrophages were significantly downregulated except for NOS2, which was upregulated in M0. ARG1 was also upregulated in M0 and M1 states following TLR4 knockdown, but the expression was unchanged in the M2 state.

### 3.4. Effects of IRF5 Knockdown on BMDM Polarization

The expression of IRF5 was moderately enhanced in M2 polarization compared with M0 and M1 (Figure 1c and Figure 4a). The potent effects of IRF5 knockdown are shown in Figure 6. The action of IRF5 siRNA was potent and consistent. Knockdown of the endogenous IRF5 message was almost complete not only in M0 and M1, where background expression was moderate, but also in M2 polarization, where it was upregulated almost 3-fold. In the M1-polarization state, knockdown of IRF5 resulted in significant downregulation of all pro-M1 markers tested: NO2, TNF-α, and Il6. Although IRF5 knockdown has no effect on ARG1 expression in the M1-polarization state, it downregulated this primary marker of M2 polarization.

If transfection with siRNA preceded LPS or IL-4 and IL-10 polarization (protocol 1), knockdown of IRF5 was effective in all macrophage states. Expression of M2-associated markers, i.e., ARG1 and EGR2, remained constant in all macrophage states except ARG1, which was significantly upregulated in M0 macrophages. As for M1-associated markers, in contrast to Figure 6, the knockdown of IRF5 upregulated the expression of NOS 2 in both M1 and M2 states: IL-6 only in the M1 state and TNF-α in the M2 state as shown if Figure 7. The contrast in results due to the different order of events may be explained by the role of epigenetics in macrophage polarization. In protocol 1, we took naïve macrophages, transfected them with siRNA, then polarized them; however, in protocol 2, we initially induced differentiation into either the M1 or M2 state, after which we transfected with siRNA on primed macrophages whose transcription factors were already poised for transcription.

### 3.5. Effect of IRF3 Knockdown on BMDM Polarization

Knockdown of IRF3 also significantly suppressed the expression of M1 proinflammatory markers NOS2, TNF-α, and Il-6 (Figure 8c–e). In contrast to IRF5, IRF3 knockdown also increased ARG1 expression in M1-polarized macrophages, making a comprehensive and complete drive towards M2 polarization. Western blot analysis of IRF3 demonstrated a long-lasting reduction in protein expression (up to 7 days of data shown in Appendix A) following the knockdown, Figure 9.

## 4. Discussion

The selection of EGR2 for targeting in macrophages was made for several reasons. EGR2 has been characterized as a potent marker for M2 polarization [16]. Recently, it was shown that this transcription factor is a master regulator in macrophages, translating transient polarization signals to stable epigenomic and transcriptional changes. We have confirmed that the expression of EGR2 is exponentially stimulated by IL-4 (plus IL-10 in our settings) (Figure 1). According to a recently published extensive investigation, EGR2 expression is activated downstream from the IL-4 receptor signalling. Transient STAT6 activation induces stable transcriptional changes. EGR2 was identified as a downstream regulator that further induces many transcription factors involved in alternative polarization. The EGR2 pathway is conserved in mouse and human alveolar macrophages, making it an attractive therapeutic target [17,18]. Complementary to the study by Veremeyko [19], we demonstrated that EGR2 knockdown enhances the M1 phenotype, as summarized in Table 1, by downregulation of ARG1 and activation of TNF-α and NOS2 in M0 and M1 macrophages (Figure 3). We observed a small effect on the expression of ARG1 in M2 macrophages by extremely high stimulation of EGR2 by IL-4 and IL-10, to a level which is not effectively controlled by RNAi, at least not at the short duration (48–72 h) which we investigated. These results are also in corroborated by the work carried out in [7], in which they silenced spliceosome-associated factor 1 (SART1) in macrophages to alleviate BLM-induced lung injury and fibrosis. The authors demonstrated that SART1, akin to EGR2, is downstream of the STAT6/PPAR-γ signalling axis, which is responsible for alternative transcriptional programming; additionally, whey showed that the suppression of SART1 in vivo inhibited the M2 macrophage program.

TLR4 is a master regulator of macrophage polarization, with its main effect being the induction of proinflammatory cytokines through the activation of the NF-κB pathway, the stimulation of NOS2 expression, and activity for efficient antimicrobial action. Despite intensive investigations of pattern-recognition receptors, among which TLR4 is the most-studied, the complexity of the pathway does not allow the prediction of in vivo effects of the inhibition of TLR4 response by RNAi, by the activation by overexpression of mRNA, or by the activation of TLR4 by its natural ligands in situ [20]. Several effects must be considered: endotoxin tolerance, back loop signalling to reduce proinflammatory and cytotoxic effects of TLR4 activation, and multiple levels of regulation of genes affected by the TLR4 pathway [21].

We would like to emphasize the potential practical importance of ARG1 downregulation by TLR4 knockdown. This effect seems unexpected, as knockdown of TLR4, an inducer of M1 polarization, leads to the downregulation of ARG1, a key enzyme of M2 polarization. However, activation of ARG1 through the TLR4 pathway is well documented and may serve as a “brake” on excessive cytotoxicity of M1, thereby thwarting NOS2 production [22]. Reducing ARG1 expression and activity may be a rational approach to counteract immunotolerant-tumour-associated macrophage effects. Our finding corroborates a recent in vivo study, which demonstrated that the tumours grew slower and the cachexia symptoms were milder in the TLR4-silenced groups. In contrast, TLR4, NOS2, IL-6, MIP-3α, and VEGF were highly expressed in the transplanted tumour tissues from the LPS groups, and their expression levels were decreased in the TLR4-silenced groups [23].

Interferon-regulatory factor 5 is a crucial transcription factor discovered during the investigation of interferon response to viral infection. Intensive research conducted for almost two decades underpinned its central role in inflammatory disorders, such as rheumatoid arthritis, inflammatory bowel disease, and systemic lupus erythematosus. IRF5 is one of the primary effectors of the TLR4/LPS signalling axis and is responsible for initiating the proinflammatory response. Most importantly, IRF5 plays a key role in determining the inflammatory phenotype of macrophages [24,25,26]. IRF5 is widely considered a potential therapeutic target [27]. RNA interference is regarded as a potential alternative to small molecule inhibitors of IRF5. Experiments in IRF5-knockout mice demonstrated that IRF5 upregulates NOS2 and Th1 responses in macrophages during *Leishmania donovani* infection [28]. IRF5^−/−^ mice show decreased type I IFN induction upon Newcastle disease virus infection [29]. IRF5 was shown to be essential in the M1 polarization of human macrophages during bacterial clearance [30]. IRF5 regulates metabolic response in alveolar macrophages with reduced ability to utilize oxidative phosphorylation, which is the most significant level of regulation [30]. M1 macrophages rely mainly on glycolysis, leading to the accumulation of microbicidal itaconate and succinate and stabilization of hypoxia-inducible factor 1α (HIF1α).

On the contrary, M2 cells are more dependent on oxidative phosphorylation [31]. Metabolic adaptations are essential to sustain macrophage polarization in specific sites of inflammation [31]. All this makes IRF5 knockdown particularly interesting for intervention in inflammatory disorders. Indeed, silencing IRF5 in vivo by intraperitoneal injection of siRNA reduced nephritis in the experimental Lupus model [32]. Additionally, it was shown that mice lacking IRF5 in myeloid cells demonstrated reduced hepatic fibrosis in the acute CCl4 toxic model and in the nonalcoholic steatosis model (NASH). It was shown that IRF5 loss of function was associated with transcriptional reprogramming of macrophages leading toward immunosuppressive and antiapoptotic properties [33]. Our data on the effective reduction in M1 polarization markers NOS2, Il6, and TNF-α make siRNA to IRF5 a strong candidate for in vivo modulation of liver fibrosis. Surprisingly, IRF5 knockdown also suppressed ARG1, the primary M2 polarization marker, in contrast to the previously published results [34].

Several investigations describe the ability of IRF3 to regulate the expression of *Ifnb1* and other individual target genes via the NF-κB pathway. These processes are related to their virus-activated transcription [35]. IRF3 was discovered as a transcription factor activating type I interferons. IRF3 is involved in antiviral and antitumour pathways. It is expressed ubiquitously in the cytoplasm and, upon activation mainly through pattern recognition receptors (PRR), dimerizes and translocates to the nucleus where it activates transcription of the genes mounting antivirus responses, particularly INF-b, Il-23, RANTES (CCL5), and IP-10 (CXCL10), as well as many others. It also can inflict apoptosis via migration to mitochondria complexing with Bax [36]. Our decision to test the effects of IRF3 knockdown on macrophage polarization was based on preceding publications demonstrating the protective effect of the global knockout of IRF3 on ethanol-induced liver fibrosis [37] and in acute or chronic CCl4-induced liver injury [38]. However, these data are contradicted by a more recent study, showing that liver injury, apoptosis, and fibrosis were enhanced in IRF3-KO mice [39]. Here, we confirm that knockdown of IRF3 has a profound effect on the expression of several polarization markers in M1- and M2-polarized macrophages. It downregulates the expression of NOS2, Il6, TNF-α, and EGR2, while enhancing ARG1 expression in the M1-polarization state, where its expression is normally suppressed.

The effect of IRF3 knockout on the expression of polarization markers was not systemically investigated; however, there were some sporadic data reported. For instance, there was no difference in the expression of NOS2 in M1 and ARG1 in M2 in macrophages isolated from the bone marrow of W.T. and IRF3-KO mice [40]. These data, however, are difficult to compare directly to the effect of siRNA knockdown on BMDM in our experiments. It might be that macrophages derived from IRF3-KO mice compensated the ability to induce NOS2 in the absence of IRF3, which are essential in W.T. macrophages for this process.

Infection of macrophages derived from IRF3-KO mice with *L. monocytogenes* failed to induce NOS2 expression observed in W.T. macrophages. However, this defect was remedied with INF beta [41]. Additionally, targeting IRF3 with shRNA prevented the production of reactive nitrogen species in response to IL-6 in Raw 264.7 cells. In IRF3^−/−^ macrophages, poly I:C failed to stimulate NO production [42]. Mechanistic experiments allowed authors to suggest that ERK and IRF3 coordinate induction of NO by macrophages in response to stimulation of TLR3 [43]. These data support our observations on the critical role of IRF3 in the induction of crucial macrophage polarization markers. Although classical sites of IRF3 binding are located in type I INF genes and RANTES, in the CBP/P300 complex, it was shown to participate in NF-κB-activated transcription as well [41]. It was suggested that interferon-stimulated gene factor 3 (ISGF3) and NF-κB indeed cooperate in producing H4 acetylation of chromatin not only in NOS2 promoter but also in other loci, thus bringing regulation on epigenetic level [41], which may explain the concerted effect of IRF3 knockdown on several different polarization markers, such as NOS2, EGR2, Il6, and TNF-α demonstrated here.

## 5. Conclusions

The plasticity of macrophages makes them pliable for reprogramming by different cytokines or procedures such as transient nucleic acid transfection to elicit specific functional phenotypes. This study investigated siRNA’s capacity to alter macrophage phenotypes by knocking down the transcription factors IRF3, IRF5, and EGR2 and the transmembrane receptor TLR4, which are all involved in the signalling pathways responsible for macrophage polarization and phenotype acquisition. The results show that EGR2 knockdown in BMDM in vitro promotes the M1 phenotype. Depending on the order of treatment, i.e., polarization first, then transfection and vice versa, IRF5 and TLR4 knockdown either potentiated or dampened the M1 phenotype, highlighting the hand of epigenetic mechanisms. IRF3 knockdown promoted M2 phenotypes. These findings lay the foundations for future experimental studies in vivo in which manipulation of macrophage phenotypes will be performed in the setting of a disease model.

## Figures and Tables

**Figure 1 cells-11-02498-f001:**
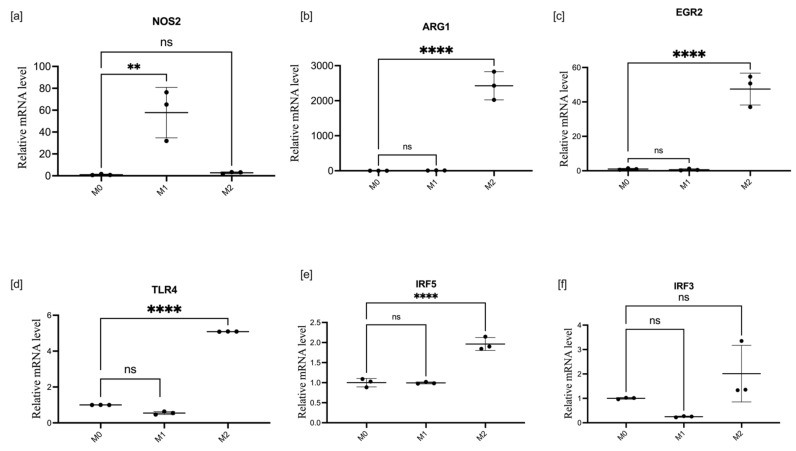
mRNA expression of target genes in different macrophage-polarization states. Relative mRNA quantity of NOS2 (**a**), ARG1 (**b**), EGR2 (**c**), TLR4 (**d**), IRF5 (**e**), and IRF3 (**f**) normalized to GAPDH expression in macrophages activated for 24 h with 100 ng/mL LPS (M1) or 20 ng/mL IL-4 plus 20 ng/mL IL-10 (M2). Data are presented as mean ± SD, *n* = 3 [ns; not significant; ** *p* < 0.01; **** *p* < 0.0001].

**Figure 2 cells-11-02498-f002:**
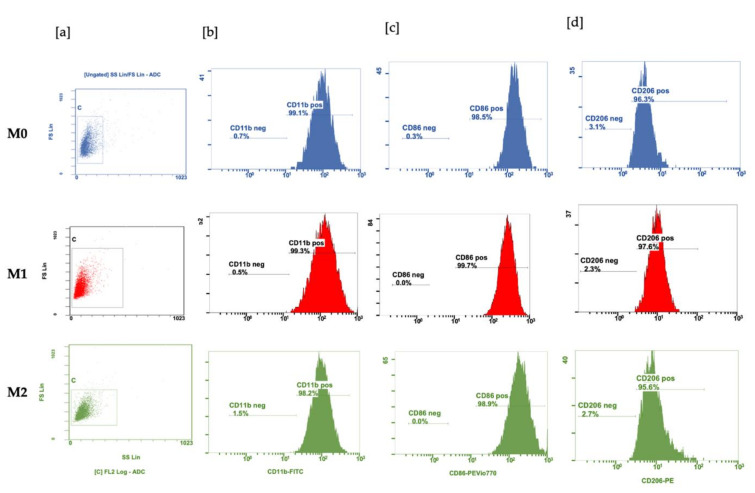
Polarization of BMDM: Macrophages were generated by incubating myeloid progenitor cells with M-CSF for seven days, then activated for 24 h with 100 ng/mL LPS (M1) or 20 ng/mL IL-4 plus 20 ng/mL IL-10 (M2). (**a**) Forward- and side-scatter profiles of macrophages and flow cytometric characterization of CD11b; (**b**) differential expression of cell surface markers CD86 (**c**) and CD206 (**d**) in M0 (blue panel), M1 (red panel) and M2 (green panel) macrophages. Based on median fluorescence intensity (MFI) *n* = 3, shown in Appendix A, the MFI CD 86 was highest in M1 as compared with M0 and M2, whilst the MFI of CD206 was highest in M2-polarized macrophages. M0 macrophages expressed intermediate amounts of surface markers. The histogram represents the results obtained from one run.

**Figure 3 cells-11-02498-f003:**
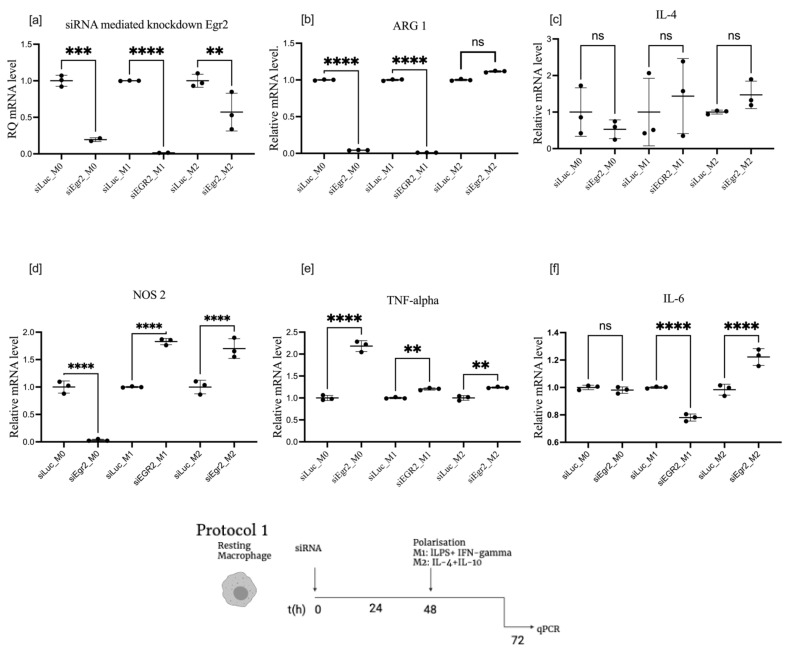
mRNA expression of EGR2 (**a**), ARG1 (**b**), IL-4 (**c**), NOS2 (**d**), TNF-α (**e**), and IL-6 (**f**) after EGR2 knockdown in post-transfected polarized BMDMs, treated according to protocol 1. Data are presented as mean ± SD, *n* = 3 [ns; not significant; ** *p* < 0.01; *** *p* < 0.001; **** *p* < 0.0001].

**Figure 4 cells-11-02498-f004:**
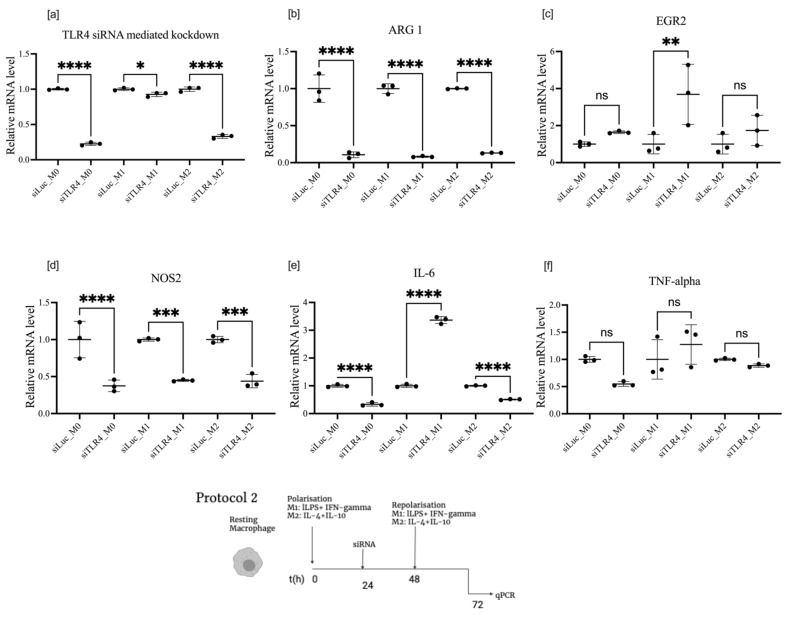
mRNA expression of TLR4 (**a**), ARG1 (**b**), EGR2 (**c**), NOS2 (**d**), IL-6 (**e**), and TNF-α
(**f**), after TLR4 knockdown in macrophages treated according to protocol 2. Data are presented as mean ± SD, *n* = 3 [ns; not significant; * *p* < 0.05; ** *p* < 0.01; *** *p* < 0.001; **** *p* < 0.0001].

**Figure 5 cells-11-02498-f005:**
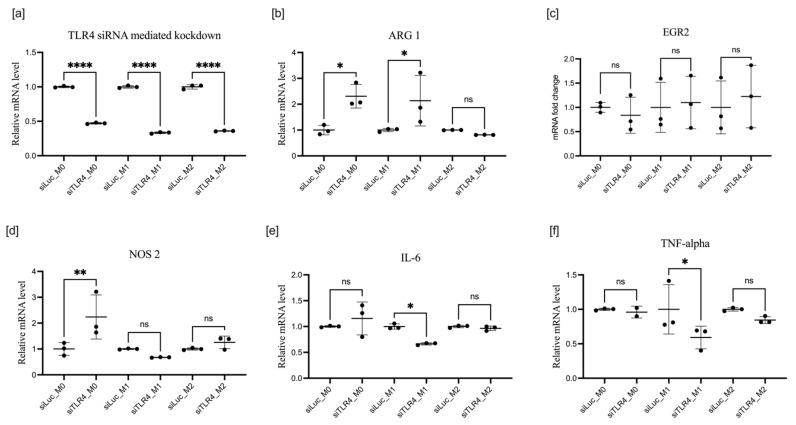
mRNA expression of TLR4 (**a**), ARG1 (**b**), EGR2 (**c**), NOS2 (**d**), IL-6 (**e**), and TNF-α (**f**) after TLR4 knockdown in macrophages treated according to protocol 1. Data are presented as mean ± SD, *n* = 3 [ns; not significant; * *p* < 0.05; ** *p* < 0.01; **** *p* < 0.0001].

**Figure 6 cells-11-02498-f006:**
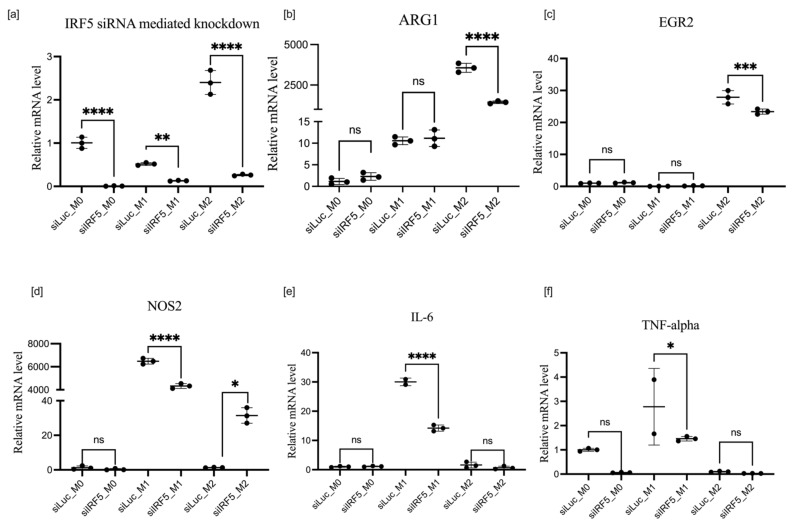
mRNA expression of IRF5 (**a**), ARG1 (**b**), EGR2 (**c**), NOS2 (**d**), IL-6 (**e**), and TNF-α (**f**) after IRF5 knockdown in macrophages treated according to protocol 2. Data are presented as mean ± SD, *n* = 3 [ns; not significant; * *p* < 0.05; ** *p* < 0.01; *** *p* < 0.001; **** *p* < 0.0001].

**Figure 7 cells-11-02498-f007:**
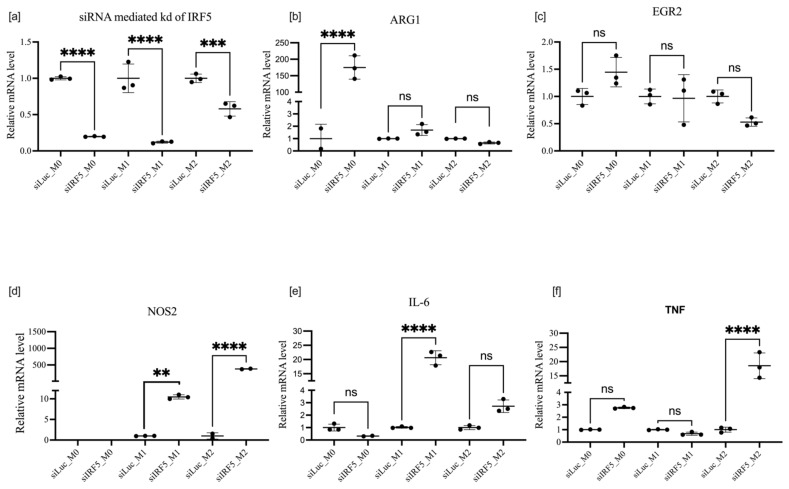
mRNA expression of IRF5 (**a**), ARG1 (**b**), EGR2 (**c**), NOS2 (**d**), IL-6 (**e**), and TNF-α  (**f**) after IRF5 knockdown in macrophages treated according to protocol 1. Data are presented as mean ± SD, *n* = 3 [ns; not significant; ** *p* < 0.01; *** *p* < 0.001; **** *p* < 0.0001].

**Figure 8 cells-11-02498-f008:**
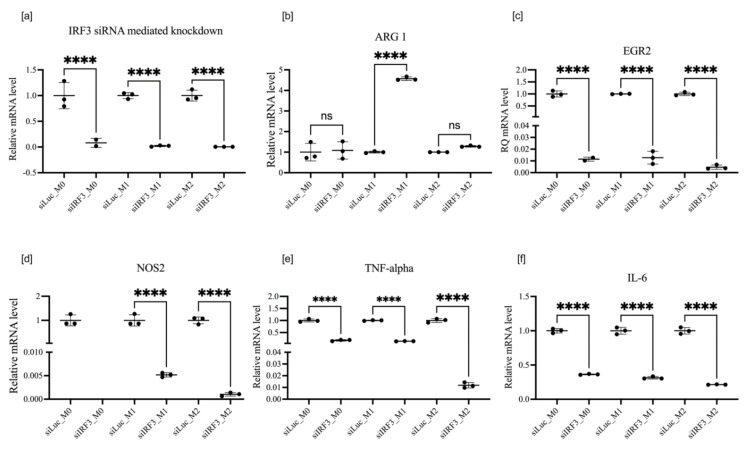
mRNA expression of IRF5 (**a**), ARG1 (**b**), EGR2 (**c**), NOS2 (**d**), IL-6 (**e**), and TNF-α  (**f**) after IRF5 knockdown in macrophages treated according to protocol 2. Data are presented as mean ± SD, *n* = 3 [ns; not significant; **** *p* < 0.0001].

**Figure 9 cells-11-02498-f009:**
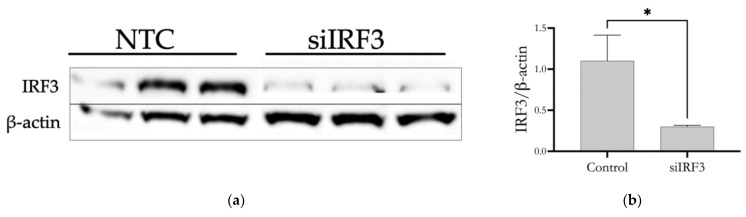
Western blot analysis of IRF3 expression in nonpolarized BMDM after siRNA-mediated knockdown of IRF3 for three days measured in triplicates (**a**). Bar graph (**b**) shows IRF3 protein quantity relative to expression of β-actin. Each bar represents the mean ± S.D. of three replicates. * *p* < 0.05. NTC indicates non-transfected control.

**Table 1 cells-11-02498-t001:** A summary of how siRNA-mediated knockdown of selected target genes alters the phenotype of BMDMs based on the sequence of transfection and polarization.

Protocol of Treatment	Target Knocked Down	Overall Impact on M1 Markers	Overall Impact on M2 Markers	Phenotype Enhanced
Transfection followed by polarization	EGR2	NOS2 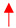	ARG1 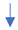	M1
TNF-α 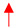	IL-4 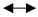
IL-6 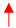	
Polarization followed by transfection	IRF3	NOS2 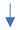	ARG1 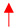	M2
TNF-α 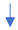	EGR2 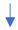
IL-6 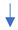	
Transfection followed bypolarization	IRF5	NOS2 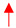	ARG1 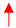	M1
TNF-α 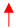	EGR2 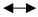
IL-6 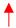	
Polarization followed by transfection	NOS2 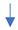	ARG1 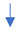	M2
TNF-α 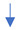	EGR2 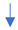
IL-6 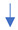	
Transfection followed bypolarization	TLR4	NOS2 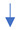	ARG1 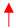	M2
TNF-α 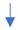	EGR2 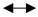
IL-6 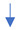	
Polarization followed by transfection	NOS2 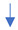	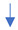 ARG1	M2
TNF-α 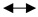	EGR2 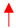
IL-6 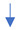	

## Data Availability

Not applicable.

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
