# Peer review of "Phenotypic Alteration of BMDM In Vitro Using Small Interfering RNA"

_cells, 2022, doi:10.3390/cells11162498_

Round 1
Reviewer 1 Report
The author addressed all my comments in the resubmitted mansucript.
Author Response
Comment 1: The author addressed all my comments in the resubmitted manuscript.
Response: Thank you very much.
Reviewer 2 Report
The authors used RNAi to knockdown EGR2, IRF3, IRF5, and TLR4, target molecules of macrophage polarization, and examined their effects on M1 and M2 markers and polarity markers. This study confirms that knockdown by RNAi is useful for detailed analysis of macrophage phenotypes. This paper has already been extensively revised by other reviewers and is well written. Therefore, I have only minor comments.
Minor points.
1. Since the manuscript is descriptive in its current state, it would be better to prepare a table summarizing the results.
2. The bands in Figure 8A needs to be improved.
Author Response
Comment 1: Since the manuscript is descriptive in its current state, it would be better to prepare a table summarizing the result
Response 1: We agree with this comment. Therefore, we have incorporated a summary table highlighting the phenotype enhanced upon knockdown of GOI as shown in the revised manuscript from pages 11-12, lines 307 to 312.
Comment 2: The bands in Figure 8A needs to be improved
Response 2: We have replaced figure 8A with a slightly better image with hopefully better bands; furthermore, the figure legend has been altered accordingly to highlight that experiment was triplicated as recommended by one of the previous reviewers.
This manuscript is a resubmission of an earlier submission. The following is a list of the peer review reports and author responses from that submission.
Round 1
Reviewer 1 Report
Halimani et al studied the evaluated and validated targets for macrophage reprogramming invitro through RNAi based polarization for prospective cell therapy. They investigated the alternative approach based on RNAi knockdown of precisely selected targets, allowing long-lasting effects.
Comments.
- Many typo errors in the manuscript eg. GPDH to GAPDH on page 3 line 104.
- Figure 1. contains confusion as page 5 part of figures marls [a]…..[f] and again page 5 starting with [a]….[d]. Please make changes to the figure.
- Is CD206 expressed at the same levels in M0, M1 and M2 ?, some error?
- Western blot analysis of IRF3 should be repeated a minimum of three times and show the quantification.
- All Figure’s legends missing formation about how many no. of experiments was performed?
- Please discuss the following paper in discussion.
Pan, T., Zhou, Q., Miao, K., Zhang, L., Wu, G., Yu, J., Xu, Y., Xiong, W., Li, Y., & Wang, Y. (2021). Suppressing Sart1 to modulate macrophage polarization by siRNA-loaded liposomes: a promising therapeutic strategy for pulmonary fibrosis. Theranostics, 11(3), 1192–1206. https://doi.org/10.7150/thno.48152.
- The manuscript requires English editing.
Reviewer 2 Report
Major concerns:
- The authors declared the RNAi based Macrophages have a prospective cell therapy, but there is no such evidence or examples presented, no macrophage transfer to cure a disease or infection for example.
- The work is not enough for this journal.
- The authors described BMDM and RAW264.7 were both used, but no this information in methods and results. Also, the results part, the authors talks about the polarization of macrophage, but RAW264.7 was used as it is already a polarized macrophage in this condition?
- Autologous macrophages transfer is a interesting topic, but instead of using RNAi to knock down the genes of macrophages, why not CRISPR/Cas9 system?
Format and other problems:
- Gene name, in vitro, ex vivo, p value et at, they should be italic written. Irregular writing, for example, EGR2 and Egr2,IL4 and Il4, IL10 and Il10, mixed using in the context. TNF-?, TNF-alpha, should be consistent
- For Qpcr, the primers used should be list at least in supplementary. For FACS study, the antibody should be declared in method part.
- In statistics part, only indicated p<0.05 as significance, but in the figures, three significance level were labeled.
- Figure 1, two parts of the results were presented, qPCR and FACS, they are separated by pages but using the same figure legend and using the repeated a b c d to label sections, there is not really a good idea.
- Figure 7, the labels above WB gels were not even at same level.
- The authors are mixing past sentence and present sentence.
- In discussion, I don’t get: (“The Metabolic Signature of Macrophage Responses”) Metabolic adaptations are essential to sustain macrophage polarization in specific sites of inflammation.
- In discussion, “Complementary to the of Veremeyko study”, bad writing or obvious mistake, this kind of carelessness appears frequently.
Reviewer 3 Report
This manuscript described a intensive study on effects of RNAi knock down of precisely selected targets EGR2, IRF3, IR5 and TLR4, on macrophage polarizations. The results shows importance of those target in macrophage polarization. Results are convincing and important in understanding macrophage polarizations. However, it may not serve as a practical method for macropharge preprograming for cell therapy, as RNAi is generally a transient effect, the phenotypes of preprogramed macrophage may subject to adaptation of the fibroinflammatory meliu of the microenvironment of diseased organ. Author may consider to change the title and context in line with actual findings by RNAis in macrophage polarization.
The language needs some improvement to clarify statements in discussion.